# Investing in Operational Research Capacity Building for Front-Line Health Workers Strengthens Countries’ Resilience to Tackling the COVID-19 Pandemic

**DOI:** 10.3390/tropicalmed5030118

**Published:** 2020-07-16

**Authors:** Rony Zachariah, Selma Dar Berger, Pruthu Thekkur, Mohammed Khogali, Karapet Davtyan, Ajay M. V. Kumar, Srinath Satyanarayana, Francis Moses, Garry Aslanyan, Abraham Aseffa, Anthony D. Harries, John C. Reeder

**Affiliations:** 1UNICEF, UNDP, World Bank, WHO Special Programme for Research and Training in Tropical Disease (TDR), 1211 Geneva, Switzerland; khogalim@who.int (M.K.); aslanyang@who.int (G.A.); armidiea@who.int (A.A.); reederj@who.int (J.C.R.); 2Center for Operational Research, International Union against Tuberculosis and Lung Disease (The Union), 75006 Paris, France; sberger@theunion.org (S.D.B.); pruthu.tk@theunion.org (P.T.); akumar@theunion.org (A.M.V.K.); adharries@theunion.org (A.D.H.); 3Center for Operational Research, The Union South-East Asia (USEA), New Delhi 110016, India; SSrinath@theunion.org; 4Country Health Policies and Systems, World Health Organization Regional Office for Europe, 2100 Copenhagen, Denmark; kdavtyan@who.int; 5Community Medicine, Yenepoya Medical College (Deemed to Be University), Yenepoya, Mangalore 575018, India; 6Reproductive Health and Family Planning Program, Ministry of Health and Sanitation, Freetown 23222, Sierra Leone; franqoline@gmail.com; 7London School of Hygiene and Tropical Medicine, Keppel Street, London WC1 7HT, UK

**Keywords:** COVID-19, operational research, health systems, SORT IT, pandemics, training

## Abstract

(1) Introduction. The Structured Operational Research and Training IniTiative (SORT IT) supports countries to build operational research capacity for improving public health. We assessed whether health workers trained through SORT IT were (1) contributing to the COVID-19 pandemic response and if so, (2) map where and how they were applying their SORT IT skills. (2) Methods. An online questionnaire survey of SORT IT alumni trained between 2009 and 2019. (3) Results. Of 895 SORT IT alumni from 93 countries, 652 (73%) responded to the survey and 417 were contributing to the COVID-19 response in 72 countries. Of those contributing, 307 (74%) were applying their SORT IT skills to tackle the pandemic in 60 countries and six continents including Africa, Asia, Europe, South Pacific and North/South America. Skills were applied to all the pillars of the emergency response with the highest proportions of alumni applying their skills in data generation/analysis/reporting (56%), situation analysis (55%) and surveillance (41%). Skills were also being used to mitigate the health system effects of COVID-19 on other diseases (27%) and in conducting research (26%). (4) Conclusion. Investing in people and in research training ahead of public health emergencies generates downstream dividends by strengthening health system resilience for tackling pandemics. It also strengthens human resources for health and the integration of research within health systems.

## 1. Introduction

“The operational research training I received from TDR and its partners has been invaluable as it has enabled me to transfer the skills I acquired while conducting research on Ebola to my current work on COVID-19”—Dr James Squire, Ministry of Health, Sierra Leone.

These words coming from a front-line doctor who led the 2014/2015 Ebola outbreak response at its epicenter in Kailahun district in Sierra Leone merit reflection. Dr Squire is now leading the Ministry of Health’s efforts to enhance surveillance systems that generate real-time, high-quality and disaggregated data for tackling Coronavirus disease 2019 (COVID-19). Encouragingly, he is applying the research skills he gained through the Structured Operational Research and Training InitiaTive (SORT IT) to his current work on COVID-19, but how exactly are these skills being applied? Such information would help inform the wider gains of investing in research training.

SORT IT is a global partnership-based initiative led by TDR, The Special Programme for Research and Training in Tropical Diseases, and implemented with various partners including ministries of health, non-governmental organizations (NGOs) and academic institutions [1]. It supports countries to build operational research capacity for strengthening health care delivery systems, improving programme performance and promoting public health [1,2]. The model is unique in that it targets front-line health workers and other programme staff, embraces “on the job” learning and simultaneously combines research training with research implementation [3].

In line with a WHO call that “all nations should be producers and consumers of research and research capacity be strengthened close to the supply of and demand for health services” [4], SORT IT has trained participants from 93 countries [5]. With 70% of research studies influencing policy and practice, SORT IT examines what works or does not work in real-world settings and introduces solutions to improve health care [6].

In the light of the COVID-19 pandemic, the link between this training programme and its role in strengthening health system resilience to respond to pandemics merits examination. We therefore assessed (1) whether SORT IT alumni are contributing to the COVID-19 pandemic response and if so, (2) map where and how they are applying their SORT IT skills.

## 2. Methods

We carried out a semi-structured questionnaire-based survey on all SORT IT alumni trained from the start of the SORT IT programme (in 2009) until December 2019. E-mails of alumni were sourced from a SORT IT web-based alumni network and a training database. Between March and April 2020, each alumnus received a SurveyMonkey link (surveymonkey.co.uk) to access the questionnaire.

The questionnaire was pre-tested and included information on demographics, whether the person was currently involved with the COVID-19 response and if so, whether he/she was applying the skills gained from the SORT IT training to the pandemic response. If the response to the latter was “yes”, the person was asked to specify the area(s) where the skills were being applied and provide some illustrative information. Up to two reminders were sent if responses were not received within 7 days. Where e-mails were invalid, social media links (Facebook Messenger, Skype and Whats App) were used to update contact details and send reminders.

The SORT IT programme covered various aspects of the research cycle such as research prioritization, formulation of the research question, study protocol writing, efficient data capture and analysis, manuscript writing and knowledge management [1]. SORT IT also has an in-built system to gather information for improving the quality and performance of the training programme [1]. Survey responders were all adults, participation was voluntary, data were anonymized, there were no personal identifiers and no sensitive personal questions were included that could risk psychological or social harm. This was thus considered a minimal risk study and the Ethics Advisory Group of the International Union Against Tuberculosis and Lung Disease, Paris, France (which oversees ethics reviews for the SORT IT global partnership), determined that ethics clearance was not required for this study. The survey data was exported to Microsoft Excel and used for data analysis.

## 3. Results

The survey covered 84 SORT IT courses with 895 alumni from 93 countries. A total of 652 (73%) alumni (female = 45%) responded to the survey (Figure 1). Of those who responded, 417 from 72 countries were actively involved in the COVID-19 response and 307 (74%) from 60 countries were applying their skills acquired from SORT IT courses to tackle the pandemic (Figure 2). The top five Low-and Middle Income-Countries (LMIC) where alumni were applying their skills included India (70), Myanmar (32), Zimbabwe (26), Kenya (19), Pakistan (18) and China (6). SORT IT Alumni were also using their acquired skills in high-income countries (HIC) including Australia, Belgium, Canada, Italy, Japan, USA, and the United Kingdom (Figure 2).

Table 1 shows various areas of the COVID-19 emergency response where SORT IT alumni were applying their skills with some illustrative quotes. Skills were being applied in all the pillars of the outbreak response, namely: situation analysis, surveillance, emergency preparedness, case management and data generation and reporting. The three areas with the highest proportion of alumni applying their skills were data generation, analysis and reporting (56%), situation analysis (55%) and surveillance (41%). Alumni were also applying their skills in mitigating the health system effects of COVID-19 on diseases such as tuberculosis, HIV and non-communicable diseases (27%) and in conducting COVID-19 related research (26%).

In total, alumni from 85 countries responded, in 72 countries they were involved in the COVID-19 response and in 60 countries they were applying their skills gained through SORT IT.

## 4. Discussion

This is one of the first studies showing that SORT IT provides skillsets and core competencies that can be used transversally in building health system resilience at the time of a pandemic. Encouragingly, about seven in every ten individuals involved with COVID-19 reported applying their SORT IT acquired skills in 60 countries, including both LMICs and HICs.

These findings show that SORT IT has equipped front-line health workers not only with research skills, but also with a skill-set needed to respond to the unprecedented COVID-19 pandemic [7]. This proves the down-stream benefits of investing in operational research capacity building. The wide geographic coverage with no dichotomy between LMICS and HICs shows that such skills are universally applicable and likely to enhance global solidarity in tackling future outbreaks and pandemics.

There might have been a perception by some donors that investing in research capacity building is a luxury that is divorced from public health action. Much funding for research training also lies with academic institutions and is not accessible to implementers from disease control programmes [3,8,9]. It is time for a *volte-face.*

The strengths of this study are that SORT IT alumni in 93 countries were contacted and specific efforts were made to validate invalid e-mails, thereby limiting non-responders. As the SORT IT programme has a robust built-in monitoring and evaluation system, we were able to make use of this existing system to gather both quantitative and qualitative information. Study limitations include a response rate of 73%, which under the circumstances is still acceptable to good, the self-reported nature of the response, the potential social desirability bias and, considering the continued expansion of the COVID-19 pandemic, possible underestimation in our figures.

There are a few other salient observations. First, skills are being applied beyond research to all the vital pillars of the outbreak response. While SORT IT teaches multiple and practical skills for activities such as generating and utilizing data, conducting operational research and using evidence to influence policy and/or practice, several transversal skills are acquired at the same time [1]. For example, skills are developed in fostering stakeholder engagement, performing situation analysis in programme settings, prioritizing health issues, ensuring quality-assured data capture and analysis, critically reviewing the scientific literature, scientific writing to the standards of a medical journal and managing knowledge. It is therefore not surprising that those who were trained through the SORT IT programme acquired a “tool-kit” of skills that can then be applied to several areas of the outbreak response.

Second, the three areas where acquired skills were particularly used were data generation, situation analysis and setting up surveillance systems. The generation of high quality, timely and disaggregated data is essential for ensuring that countries tackling COVID-19 become “data rich, information rich and action rich”—a fundamental goal of the SORT IT programme [2]. Conducting a sound situation analysis and setting up robust surveillance systems are crucial in any outbreak: these help to feel and monitor the pulse of an outbreak and prevent responders from thinking and acting blindly.

Third, with the lock-down and restricted movements imposed by COVID-19, individuals with chronic diseases such as tuberculosis, HIV/AIDS and non-communicable diseases will understandably face hurdles in accessing diagnostic and treatment facilities and adhering to follow-up schedules [10]. It is encouraging that SORT IT alumni were using their skills in offsetting these negative health system effects.

Finally, following the 2014/2015 Ebola outbreak, WHO spearheaded global efforts to avert epidemics by making Research and Development (R&D) “outbreak-ready” [7]. While this will accelerate R&D on vaccines, drugs, and diagnostics, finding out “how to deliver” these innovations in an equitable manner is imperative [8]. SORT IT could play an important role in such operational research.

In conclusion, the results of this study demonstrate the value of investing in people and in research training ahead of public health emergencies. Clearly, building upstream operational research capacity has generated downstream dividends in strengthening health system resilience for tackling pandemics. In addition, it strengthens human resources for health (HRH) and the integration of research within health systems. In summary, it allows the health system to have the right people in the right place at the right time.

## Figures and Tables

**Figure 1 tropicalmed-05-00118-f001:**
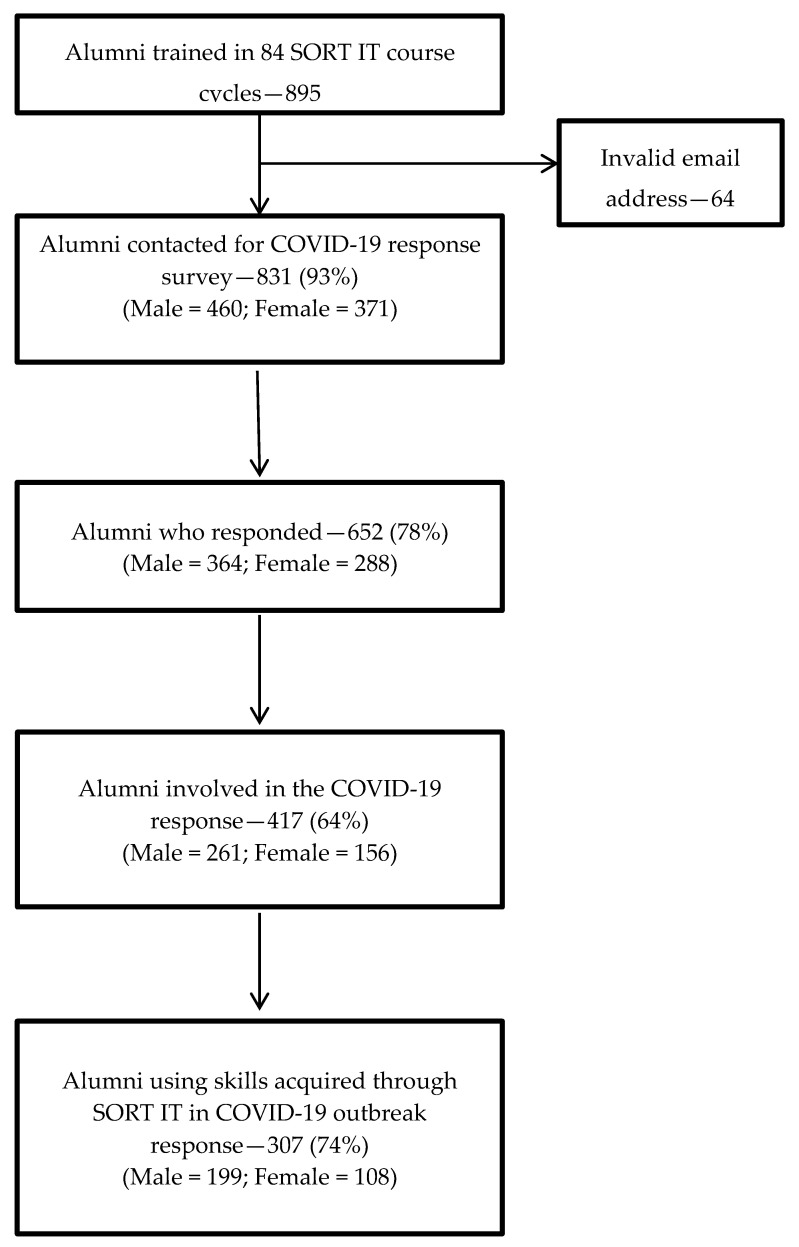
Flowchart showing response rate, involvement in COVID-19 and utilization of skills acquired through the Structured Operational Research and Training InitiaTive (SORT IT, 2009–2019).

**Figure 2 tropicalmed-05-00118-f002:**
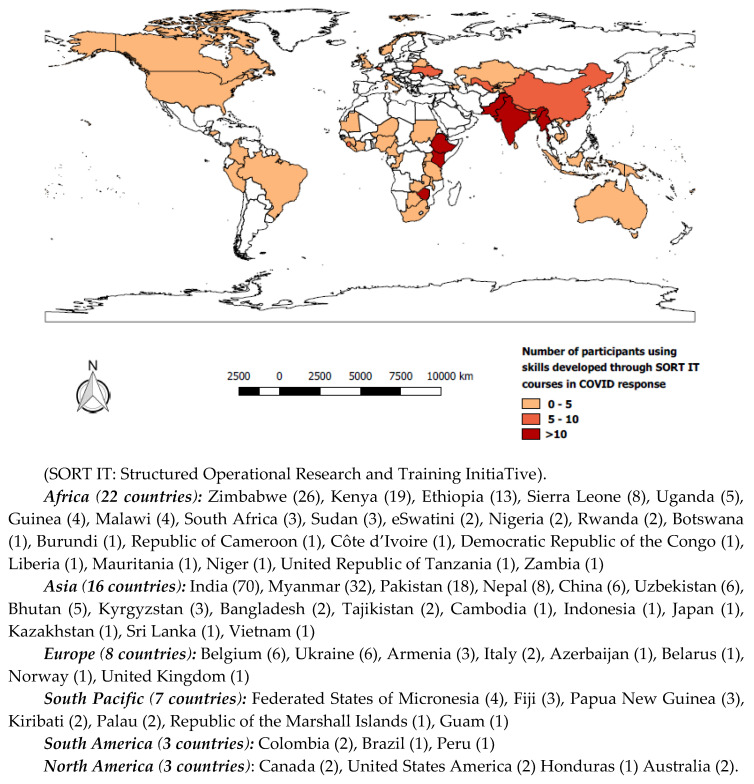
Geographic mapping of 307 SORT IT alumni applying their acquired SORT IT skills in 60 countries to tackle the COVID-19 pandemic.

**Table 1 tropicalmed-05-00118-t001:** Categories of the COVID-19 pandemic response where trainees applied skills gained through the Structured Operational Research and Training InitiaTive (SORT IT, 2009–2019). Illustrative quotes are provided for each category (N-307).

Categories of the COVID-19 Response	Applying SORT IT Skills n (%)	Illustrative Quotes from Trainees around the Globe
Situation analysis	168 (55)	I was able to conduct a situation analysis on the Taftan border of Baluchistan and establish measures for screening, management and reporting of COVID-19—PakistanI conducted a situational analysis in preparedness of health facilities to respond to COVID-19. I looked at availability of IPC equipment/supplies, staff capacity in case identification and management, and IPC practices to prevention spread—Uganda
Epidemic surveillance	126 (41)	I helped introduce a mobile health surveillance tool (TeCHO+) that is being used by community health workers to screen over 30 million people for COVID-19 in Gujarat—IndiaI was able to adapt the District Health Information System (DHIS2) to set up a surveillance and tracking system in my country—Bhutan
Emergency preparedness and response	118 (38)	I am part of the rapid investigation team of MoH/WHO for outbreak investigation, emergency preparedness and response for Rohingya refugees in Cox’s Bazar—BangladeshThe SORT IT knowledge helped me develop a mixed method study on health system preparedness to respond to the COVID-19 pandemic. It also helped my team to have an analysis plan to improve understanding of the findings by policy makers—Guinea
Infection, prevention and control (IPC) including health worker safety	145 (47)	The knowledge I acquired from the SORT IT training on Infection, Prevention and Control is being applied to the COVID-19 response—ZambiaI was able to conduct a survey on the status of preparedness of health care workers for COVID-19—Kazakhstan
Clinical management (screening, diagnosis and clinical care)	82 (27)	I am leading a team in 3 districts Hospitals and 42 health centers on screening, protocol development and trainings—RwandaI reviewed literature, analyzed data from quarantine centers and appraised the Ministry of Health and the Prime Minister’s office on decisions regarding duration of quarantine and follow up thereafter—Bhutan
Data generation analysis and reporting	173(56)	I was able to do data analysis of all surveillance reports received from the 35 facilities in a sub-county—KenyaI am developing a mobile reporting application for suspected COVID-19 patients. The code-books learnt from SORT IT are very useful for the software development—Myanmar
Operational or clinical research	79 (26)	My team are helping projects in different parts of the world to develop “simple” tools to efficiently capture relevant COVID-19 data to be used for operational research—Médecins Sans Frontierès—BelgiumAs a research analyst for homeless shelters, the data analysis and writing skills I learnt, and the application of operational research principles are critical for my current work—Canada
Mitigating COVID-19 effect on other diseases (TB, HIV/AIDS, NCDs)	83 (27)	I was able to use routine programme data to highlight significant declines in uptake of routine antenatal services and specific measures are being taken to address this in the community and at health facilities—Sierra LeoneI was able to instruct health staff in the endocrine and diabetic clinics on Infection, Prevention and Control measures and re-arranged scheduling to reduce health worker exposure to COVID-19—Sri Lanka
Others	23 (8)	*I am better at thinking more logically which is useful in all that I do—United Kingdom* *I was able to organize courses, seminars and meetings with health authorities and to prepare flow charts for patient care—Honduras*

TB: Tuberculosis; HIV/AIDS: Human Immune Deficiency Virus/Acquired Immune Deficiency Syndrome; NCDS: Non-Communicable Diseases. Many participants reported using several skills, and hence numbers and percentages are more than 488 and 100% respectively.

## Data Availability

De-identified study data are available on reasonable request from the corresponding author (zachariahr@who.int). A justification for its further use should be provided.

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
