# Peer review of "Investing in Operational Research Capacity Building for Front-Line Health Workers Strengthens Countries’ Resilience to Tackling the COVID-19 Pandemic"

_tropicalmed, 2020, doi:10.3390/tropicalmed5030118_

Round 1

Reviewer 1 Report

This is a well written report of a questionnaire sent to alumni of the SORT IT courseto improve data collection and applied scientific analysis and follow-up for health care managers of all levels across health care systems in low, middle and high-income countries. 

Major points: none.

Minor points:

1) Although the authors mention the response rate of 73% a potential weakness, I would argue that such response rate is unusually high for questionnaires in general. A potential weakness therefore is not the response rate per se (which is very high) but the potential bias of responses among those that appreciated the course, or those that feel obliged to provide socially desirable answers . .

2) I am not sure whether lines 123-126 should be maintained. Even though I would intuitively agree witrh the authors that training in research skills is highly desirable to improve health care systems and pandemic preparedness, the conclusion is the opinion of the authors that all have a keen interest in the success and future expansion of their training program. The lines are not based on the data presented, ans such lines could be in an accompanying Comment or Editorial, but these lines are a bit over the top; I suggest to leave it out.   

Author Response

POINT BY POINT RESPONSES TO THE REVIEWER COMMENTS

Tropical Med – 464624

Response to Reviewer 1

Thank you very much for reviewing this paper and your useful comments and suggestions. We have revised the manuscript in line with your suggestions. We have provided a point-by-point response to your comments and suggestions using bold font.

Reviewer: This is a well written report of a questionnaire sent to alumni of the SORT IT course to improve data collection and applied scientific analysis and follow-up for health care managers of all levels across health care systems in low, middle and high-income countries. 

Response: Thank you

Major points: none.

Minor points:

Reviewer: 1) Although the authors mention the response rate of 73% a potential weakness, I would argue that such response rate is unusually high for questionnaires in general. A potential weakness therefore is not the response rate per se (which is very high) but the potential bias of responses among those that appreciated the course, or those that feel obliged to provide socially desirable answers.

Response: We agree and have highlighted social desirability bias as one of the limitations (lines 135-138)

Reviewer: 2) I am not sure whether lines 123-126 should be maintained. Even though I would intuitively agree witrh the authors that training in research skills is highly desirable to improve health care systems and pandemic preparedness, the conclusion is the opinion of the authors that all have a keen interest in the success and future expansion of their training program. The lines are not based on the data presented, ans such lines could be in an accompanying Comment or Editorial, but these lines are a bit over the top; I suggest to leave it out.   

Response: We have rephrased this paragraph but prefer to maintain it, albeit in a rephrased form in order to advocate for increased investment in research training (see current lines 128-130). The data supports this appeal. We thank you for your kind understanding.  

Reviewer 2 Report

Congratulations to the authors for this manuscript that evaluates the contribution of capacity building received during the SORT-IT program on the COVID-19 response. A further commendation for this author group on representation from high and LMICs.  I do make the observation of gender imbalance and request that the group reflect on the reasons for this and seek to address in the future (recognising there is no short term solution - see comment 2 below) 

General Comments 

(1) Overall conclusion (title, abstract) - strengthening heath system resilience.  

Whilst I agree that the end outcome of SORT-IT and capacity building programs is strengthening the health system, there  are  at least 2 intermediate outcomes -  strengthening human resources for health (HRH) & integration of research within the health system. The last point (research) is an essential and critical investment for improved epidemic response capability and historically overlooked as and "additional measure" if resources permit.  For COVID-19, an effective global will entirely rely on research in LMICs, specifically operational / implementation research to increase the uptake of new tools as they are developed. You've made this point well in line 123

I would suggest that the authors consider adding specifics outcomes of HRH and research in the conclusion in abstract and manuscript  (lines 34, 71, 163)

2.  Results - characteristics of respondents 

a) Is there a previous publication (or reference / appendix) that details the characteristics of all the SROT-IT participants in this period.  It would be interesting to compare this to the respondents and note if there were any differences, reflecting non-response bias?  Noting that the response rate of 73% in these circumstances is very good. 

b) Can you please disaggregate the respondents and according to gender?  This is such a key metric in global health. 

c) Is it known what role the respondents had e.g clinician, researcher, public health? 

d) Can the distribution of respondents (line 100, figure 2) be compared to the geographic distribution of all course participants and be provided in proportions (or at least comment on?)  Interested to know if there was a non-response bias e.g. due to internet availability or factors such an active COVID epidemic in March-April.  

Specific comments 

Lines 42, 63 - Does SORTIT also focus on the public health workforce (which some may not consider "front-line") 

Figure 2 -  check map. I focussed on the Pacific countries as they are not visible. PNG is stated as 3 participants, but coloured white.  

Author Response

Response to Reviewer 2

Thank you very much for reviewing this paper and your useful comments. We have revised the manuscript in line with your suggestions. We have provided a point-by-point response to your comments and suggestions using bold font.

Reviewer: Congratulations to the authors for this manuscript that evaluates the contribution of capacity building received during the SORT-IT program on the COVID-19 response. A further commendation for this author group on representation from high and LMICs.  I do make the observation of gender imbalance and request that the group reflect on the reasons for this and seek to address in the future (recognizing there is no short term solution - see comment 2 below) 

Response: Thank you. Female gender constitutes about 45% of our SORT IT participants and we are trying to increase this to 50%. We do appreciate the suggestion of the reviewer to reflect on reasons for the lower proportion of women and will consider this in future research.

General Comments 

Reviewer: (1) Overall conclusion (title, abstract) - strengthening heath system resilience.  

Whilst I agree that the end outcome of SORT-IT and capacity building programs is strengthening the health system, there are  at least 2 intermediate outcomes - strengthening human resources for health (HRH) & integration of research within the health system. The last point (research) is an essential and critical investment for improved epidemic response capability and historically overlooked as an "additional measure" if resources permit.  For COVID-19, an effective global will entirely rely on research in LMICs, specifically operational / implementation research to increase the uptake of new tools as they are developed. You've made this point well in line 123

I would suggest that the authors consider adding specifics outcomes of HRH and research in the conclusion in abstract and manuscript (lines 34, 71, 163)

Response: This is a great suggestion. We have added the intermediate outcomes to the conclusions in both the abstract and main manuscript. (Current lines 35 and 169)

Reviewer:2.  Results - characteristics of respondents 

  1. Is there a previous publication (or reference / appendix) that details the characteristics of all the SORT-IT participants in this period.  It would be interesting to compare this to the respondents and note if there were any differences, reflecting non-response bias?  Noting that the response rate of 73% in these circumstances is very good. 

Response: We unfortunately do not have another publication covering the same period and thus are unable to comment further on this. We also did not collect data on non-responders but will consider this in future research. We thank you for your understanding.

Reviewer: b) Can you please disaggregate the respondents and according to gender?  This is such a key metric in global health. 

Response: We have provided the disaggregation below and have now included the gender breakdown in Figure 1 (line 176)

Table showing gender-wise analysis of response rate, involvement in COVID and utilization of skills developed through SORT IT course among alumni trained in SORT IT course during 2009-2019

Particulars

Male

Female

n

(%)

n

(%)

Alumni trained through SORT IT

493

402

Alumni with valid email address and contacted for COVID response survey

460

(93)

371

(92)

Alumni responded to COVID response survey

364

(79)

288

(72)

Alumni involved in COVID response activity

261

(72)

156

(54)

Alumni using skill developed through SORT IT for COVID outbreak response

199

(76)

108

(69)

Reviewer: c) Is it known what role the respondents had e.g clinician, researcher, public health? 

Response: Unfortunately, since the survey included busy front-line workers, we avoided going into such details as we tried to keep the survey as simple as possible to maximize the response rate.

Reviewer: d) Can the distribution of respondents (line 100, figure 2) be compared to the geographic distribution of all course participants and be provided in proportions (or at least comment on?)  Interested to know if there was a non-response bias e.g. due to internet availability or factors such an active COVID epidemic in March-April.  

Response: Unfortunately, the survey did not collect details on non-responders nor on availability of internet etc. as we endeavored to keep the survey simple. However, we will consider this in future research and thank you for your kind understanding.

Specific comments 

Reviewer: Lines 42, 63 - Does SORTIT also focus on the public health workforce (which some may not consider "front-line") 

Response: Yes. We have now mentioned this in lines 43 and 66.

Reviewer: Figure 2 -  check map. I focussed on the Pacific countries as they are not visible. PNG is stated as 3 participants, but coloured white.  

Response: Thank you. There was indeed an error which has now been corrected (Figure 2)

Reviewer 3 Report

This is a well set out paper and easy to digest. However, I would suggest that it would help the reader better assess the findings if the authors would add a short section that sets out, at a high level, the subject headings/topics covered in the SORT IT training.

There are a couple of minor typos to be addressed eg line 154....

The findings of this study are clearly relevant, particularly at this time. 

Author Response

Response to Reviewer 3

Thank you very much for reviewing this paper and your useful comments. We have revised the manuscript in line with your suggestions. We have provided a point-by-point response to your comments and suggestions using bold font.

Reviewer: This is a well set out paper and easy to digest. However, I would suggest that it would help the reader better assess the findings if the authors would add a short section that sets out, at a high level, the subject headings/topics covered in the SORT IT training.

Response: Thank you. As suggested, we have added a short description in the Methods section (lines 90-92).

Reviewer: There are a couple of minor typos to be addressed eg line 154....

Response: Corrected (line 159) and a spell check has now been done on the whole document.

Reviewer: The findings of this study are clearly relevant, particularly at this time. 

Response: Thank you.